# Correlation between PD-L1 Expression of Non-Small Cell Lung Cancer and Data from IVIM-DWI Acquired during Magnetic Resonance of the Thorax: Preliminary Results

**DOI:** 10.3390/cancers14225634

**Published:** 2022-11-16

**Authors:** Chandra Bortolotto, Giulia Maria Stella, Gaia Messana, Antonio Lo Tito, Chiara Podrecca, Giovanna Nicora, Riccardo Bellazzi, Alessia Gerbasi, Francesco Agustoni, Robert Grimm, Domenico Zacà, Andrea Riccardo Filippi, Olivia Maria Bottinelli, Lorenzo Preda

**Affiliations:** 1Diagnostic Imaging and Radiotherapy Unit, Department of Clinical, Surgical, Diagnostic, and Pediatric Sciences, University of Pavia, 27100 Pavia, Italy; 2Radiology Institute, Fondazione IRCCS Policlinico San Matteo, 27100 Pavia, Italy; 3Unit of Respiratory Diseases, Department of Medical Sciences and Infective Diseases, Fondazione IRCCS Policlinico San Matteo, 27100 Pavia, Italy; 4Department of Internal Medicine and Medical Therapeutics, University of Pavia, 27100 Pavia, Italy; 5Department of Electrical, Computer and Biomedical Engineering, University of Pavia, 27100 Pavia, Italy; 6Department of Medical Oncology, Fondazione IRCCS Policlinico San Matteo, 27100 Pavia, Italy; 7MR Application Predevelopment, Siemens Healthcare GmbH, 91052 Erlangen, Germany; 8Siemens Healthcare, 20128 Milano, Italy; 9Department of Radiation Oncology, Fondazione IRCCS Policlinico San Matteo, 27100 Pavia, Italy

**Keywords:** non-small cell lung cancer (NSCLC), programmed death-ligand 1 (PD-L1), intravoxel incoherent motion (IVIM)

## Abstract

**Simple Summary:**

Intravoxel incoherent motion diffusion-weighted imaging (IVIM-DWI) is an advanced magnetic resonance imaging (MRI) technique able to distinguish true diffusion from microcirculation-related perfusion without the use of contrast medium. Immunohistochemical analysis is the gold standard method to assess the programmed death-ligand 1 protein (PD-L1) expression status in patients affected by non-small cell lung cancer (NSCLC) to guide immunotherapy. We present our preliminary results on the evaluation of IVIM-DWI parameters and their correlation with the PD-L1 expression status in patients affected by stage III NSCLC. Since PD-L1 expression is very heterogeneous in NSCLCs, and an invasive biopsy of the tumor is necessary for immunohistochemical analysis, a non-invasive alternative method to quantify PD-L1 expression should be considered to provide information on the whole tumor. In the future, IVIM-DWI parameters could offer the possibility to perform diagnosis, pathological classification, to guide therapy, and to assess therapeutic responses.

**Abstract:**

This study aims to investigate the correlation between intravoxel incoherent motion diffusion-weighted imaging (IVIM-DWI) parameters in magnetic resonance imaging (MRI) and programmed death-ligand 1 (PD-L1) expression in non-small cell lung cancer (NSCLC). Twenty-one patients diagnosed with stage III NSCLC from April 2021 to April 2022 were included. The tumors were distinguished into two groups: no PD-L1 expression (<1%), and positive PD-L1 expression (≥1%). Conventional MRI and IVIM-DWI sequences were acquired with a 1.5-T system. Both fixed-size ROIs and freehand segmentations of the tumors were evaluated, and the data were analyzed through a software using four different algorithms. The diffusion (D), pseudodiffusion (D*), and perfusion fraction (pf) were obtained. The correlation between IVIM parameters and PD-L1 expression was studied with Pearson correlation coefficient. The Wilcoxon–Mann–Whitney test was used to study IVIM parameter distributions in the two groups. Twelve patients (57%) had PD-L1 ≥1%, and 9 (43%) <1%. There was a statistically significant correlation between D* values and PD-L1 expression in images analyzed with algorithm 0, for fixed-size ROIs (189.2 ± 65.709 µm²/s × 10^4^ in no PD-L1 expression vs. 122.0 ± 31.306 µm²/s × 10^4^ in positive PD-L1 expression, *p* = 0.008). The values obtained with algorithms 1, 2, and 3 were not significantly different between the groups. The IVIM-DWI MRI parameter D* can reflect PD-L1 expression in NSCLC.

## 1. Introduction

Lung cancer is the leading cause of cancer incidence and mortality worldwide, accounting for an estimated 2 million diagnoses and 1.8 million deaths per year [1]. The World Health Organization (WHO) classification system divides epithelial lung tumors into two major cell types: small cell lung cancer (SCLC) and non-small cell lung cancer (NSCLC) [2]. In advanced-stage patients with NSCLC, for whom surgery is not indicated, the availability of predictive biomarkers for target molecular therapy or immunotherapy has opened new treatment possibilities in addition to conventional chemo-radiotherapy.

Immunotherapy has emerged as a promising therapeutic strategy for NSCLC both in locally advanced and metastatic settings. To evade host immune surveillance, cancer cells can inhibit the immune system through inhibitory pathways such as cytotoxic T-lymphocyte-associated protein 4 (CTLA-4) or programmed-cell death 1 (PD-1) and its ligand (PD-L1). PD-L1 is a transmembrane protein which can be expressed by cancer cells, and its binding with PD-1, expressed by lymphocytes, causes downregulation of the T-cell response, directly suppressing the endogenous anti-tumor cytolytic T-cell activity. Monoclonal antibodies that block the interaction between PD-1 and PD-L1 abrogate the immune tolerance exerted by tumors through the PD-1/PD-L1 pathway.

Currently, six immune checkpoint inhibitors (atezolizumab, cemiplimab, durvalumab, ipilimumab, nivolumab, and pembrolizumab) have been approved by the United States (US) Food and Drug Administration (FDA) for a variety of indications following the publication of clinical trials demonstrating their efficacy in improving therapeutic response in patients with NSCLC. In the latest report, overall survival (OS) outcomes favored pembrolizumab versus chemotherapy as first-line therapy regardless of the PD-L1 tumor proportion score (TPS) (hazard ratio for TPS ≥ 50%, 0.68; TPS ≥ 20%, 0.75; TPS ≥ 1%, 0.79), with estimated 5-year OS rates with pembrolizumab of 21.9%, 19.4%, and 16.6%, respectively, compared to an OS of 5.5% achieved with cytotoxic chemotherapies before the advent of immunotherapy [3]. Moreover, the efficacy of immunotherapy in combination with platinum-based chemotherapy versus chemotherapy alone was demonstrated in two phase III randomized trials in patients with metastatic NSCLC, regardless of PD-L1 expression levels [4,5].

With immunohistochemical analysis, the expression of the PD-L1 by the tumor cells is quantified. However, a biopsy of the tumor is necessary, which is an invasive diagnostic method that only samples a part of the tumor. For these reasons, it is of great importance to find a non-invasive alternative way to quantify PD-L1 expression, which could possibly provide us with information on the whole tumor rather than on a small sample of it.

Recently, the association between PD-L1 expression status in lung adenocarcinoma and spectral computed tomography (CT) imaging parameters has been studied [6], as well as the metabolic parameters of fluorodeoxyglucose positron-emission tomography (FDG PET/CT) [7].

The application of magnetic resonance imaging (MRI) to the lung has been a challenge for a long time due to the characteristics of the organ in question, which do not adapt well to the operating principles of this imaging modality. In particular, the main problems were low proton density, low signal-to-noise ratio (SNR), and movement artifacts. In the last few years, MRI of the lungs has gained growing interest, thanks to the development of MRI hardware and rapid imaging technologies, such as improved gradient performance and free-breathing acquisition [8,9]. MRI does not require the use of ionizing radiation and it can provide not only morphologic information but also better tissue characterization with respect to CT [10]. Intravoxel incoherent motion diffusion-weighted imaging (IVIM-DWI) is an advanced MRI technique potentially able to distinguish true diffusion from microcirculation-related perfusion without the use of contrast medium [11]. With IVIM-DWI, it is possible to extract the pure diffusion coefficient (D), the pseudodiffusion coefficient (D*), and the perfusion fraction (pf) parameters through a biexponential analysis: D quantifies the true diffusion of water molecules in the extracellular space, and it is affected by tissue microstructure, D* describes the collective motion of blood by quantifying the incoherent motion of water molecules within the capillary network, and pf quantifies the perfusion fraction. In this study, we evaluated the correlation between PD-L1 expression in NSCLCs and their IVIM-DWI parameters, and, as secondary endpoint, we assessed the performance of different encoding systems and segmentation methods.

## 2. Materials and Methods

From April 2021 to April 2022, we prospectively enrolled a consecutive group of patients diagnosed with locally advanced NSCLC (according to TNM classification VIII edition, stage III A-C cancer and T ≥ 2); all of them had previously undergone complete tumor staging and pathological characterization as well as immunohistochemistry assay for PD-L1 quantification. The following patients were excluded: (I) patients who had no adequate compliance capabilities and characteristics to guarantee the correct execution of the MRI examination (e.g., no claustrophobia, no contraindications to MRI such as pacemakers); (II) those who had received treatment before MRI; (III) patients with lung tumors not classified as NSCLC, and (IV) those who were not tested for PD-L1 expression. PD-L1 TPS was quantified through the 22C3 pharmDx assay (Agilent Technologies, Carpinteria, CA) (Figure 1). Based on immunohistochemistry (IHC) results, the tumors were distinguished into two groups: no PD-L1 expression (<1%), and positive PD-L1 expression (≥1%). This study was conducted in accordance with the ethical standards laid down in the Declaration of Helsinki and was approved by the Ethics Committee of the hospital.

All patients underwent bilateral MRI lung examination. Conventional MRI and IVIM-DWI were performed with a 1.5-T system (MAGNETOM Aera; Siemens Healthcare, Erlangen, Germany) using a 32-channel surface coil. During the examination, both free-breathing and breath-hold sequences were used. Plain scan sequences included: a cross-sectional and coronal volumetric interpolated breath-hold examination (VIBE) T1 sequence, axial IVIM-DWI, and functional sequences of ventilation. For the axial VIBE T1 scan, the matrix was 320 × 320, the repetition time (TR) was 2.12 ms, the echo time (TE) was 0.73 ms, the field of view (FOV) was 450 mm, the layer thickness was 2.5 mm, the layer spacing was 0 mm, and the number of layers was 60. For axial IVIM, the parameters were as follows: b values 0, 10, 20, 30, 50, 70, 100, 150, 200, 400, and 800 s/mm^2^, TR 4800 ms, TE 64 ms, slice thickness 6 mm, interval 1,2 mm, matrix 160 × 160, FOV 40 × 40 cm, and number of slices = 30.

The acquired images were analyzed and post-processed by a prototype research software (MR Body Diffusion Toolbox V1.3.0., Siemens Healthcare GmbH, Erlangen, Germany).

Six circular regions of interest (ROIs) with a fixed size of 8 mm diameter were manually drawn on the tumoral mass without an overlap between them: four ROIs located in the slice where the tumor was bigger (Figure 2A), and two regions that were respectively superior and inferior to that slice. Moreover, a freehand ROI was manually drawn to include the whole extent of the lesion and avoid the surrounding normal lung tissue (Figure 2B). In the case in which the tumoral mass was too small to include the six fixed-size ROIs, only a single ROI was placed.

The ROIs data were then analyzed by the software using four different prototype algorithms: algorithm 0, characterized by a linear fitting of D, D* and pf; algorithm 1, characterized by a non-linear fitting of D*; algorithm 2, which ignores the D contribution for low b-values, with a linear fitting of all parameters; algorithm 3 characterized by a full non-linear fitting of all parameters (Table 1).

In the end, IVIM-DWI parameters including D, D* and pf values were extracted and measured through a biexponential fit of signal intensity. Instead, the apparent diffusion coefficient (ADC) was computed by conventional log-linear regression of the mono-exponential signal model, using all the acquired b-values, with the following equation: ln(S/S_0_) = −ADC × b, where S represents the signal intensity at a specified b value, and S_0_ is the signal intensity at b = 0 s/mm^2^. The ADC measures the magnitude of diffusion of water molecules within tissues, making it an indicator of cellularity; indeed, tissues with high cell density exhibit lower ADC values than those with low cell density.

Matlab R2021b was used for preprocessing patients’ data and for correlation analysis and testing. For each patient, 22 parameters have been computed by the four different algorithms of the MR Body Diffusion Toolbox on the selected regions of interest (ROIs). Pearson correlation analysis was used to analyze the linear relation between the IVIM variables and the PD-L1 expression status. The Wilcoxon–Mann–Whitney test was performed to find significant differences (*p*-value < 0.05) between the two levels of expression of the PD-L1 protein. Significant results are reported along with the related false discovery rate. Boxplots were used for visual representation and to verify the consistency with the correlation analysis.

## 3. Results

### 3.1. Patients’ Clinical Data

A total of 21 patients were included in the study. Among them, 15 (71%) were males and 6 (29%) were females. All the patients were of Caucasian ethnicity, aged between 48 and 83, with an average age of 66. There were 7 patients (33%) with lung squamocellular carcinoma, 7 patients (33%) with lung adenocarcinoma, and 7 patients (33%) with poorly differentiated NSCLC. Of these, 12 (57%) had a positive PD-L1 expression (≥1%), while 9 (43%) had no PD-L1 expression (<1%).

### 3.2. Correlation of ADC, D, D*, and pf Values with PD-L1 Expression in NSCLC

For images analyzed with algorithm 0, and for tumor ROIs segmented by fixed-size ROIs, the D values of the IVIM parameter were 1173.3 ± 289.541 µm²/s × 10^6^ for the group with no PD-L1 expression (Group 0) and 1327.0 ± 372.965 µm²/s × 10^6^ for the group with positive PD-L1 expression (Group 1) (*p* = 0.356). The D* values were 189.2 ± 65.709 µm²/s × 10^4^ for Group 0 and 122.0 ± 31.306 µm²/s × 10^4^ for Group 1 (*p* = 0.008). The pf values were 274.278 ± 184.947 µm²/s × 10^3^ for Group 0 and 209.660 ± 82.112 µm²/s × 10^3^ for Group 1 (*p* = 0.549). The ADC values were 1445.8 ± 359.581 µm²/s × 10^6^ for Group 0 and 1519.8 ± 310.672 µm²/s × 10^6^ for Group 1 (*p* = 0.842). For the same images analyzed with algorithm 0, but for freehand-drawn tumor ROIs, the D values of the IVIM parameter were 1245.5 ± 348.784 µm²/s × 10^6^ for Group 0 and 1360.6 ± 368.465 µm²/s × 10^6^ for Group 1 (*p* = 0.804). The D* values were 174.789 ± 79.057 µm²/s × 10^4^ for Group 0 and 139.625 ± 33.085 µm²/s × 10^4^ for Group 1 (*p* = 0.145). The pf values were 266.811 ± 183.11 µm²/s × 10^3^ for Group 0 and 264.092 ± 136.647 µm²/s × 10^3^ for Group 1 (*p* = 0.804). The ADC values were 1495.5 ± 334.843 µm²/s × 10^6^ for Group 0 and 1630.8 ± 395.102 µm²/s × 10^6^ for Group 1 (*p* = 0.594).

For the correlation analysis, the target value of PD-L1 expression was quantified in terms of the percentage of expression. For the fixed-size ROIs, the highest correlation value in absolute terms, equal to −0.374, related the attribute D* (IVIM DP Mean) to the expression of PD-L1, specifically for the images analyzed with the algorithm 0. Parameter ADC (ADC Mean) did not show a significant correlation with the PD-L1 target outcome, as well as D (IVIM D Mean) and pf (IVIM FP Mean) (Appendix A). For the freehand ROIs, the correlation with D* was −0.449 for algorithm 1, and −0.386 for algorithm 0. For fixed-size ROIs, ADC, D, and pf did not show any significant correlation (Appendix A).

From the Wilcoxon-Mann-Whitney (WMW) test on the fixed-size ROIs, D* had a *p*-value of 0.008 for the algorithm 0 (Figure 3). The false discovery rate for this parameter corresponded to 16%. Instead, ADC, D, and pf had no significant differences in the groups for any of the four algorithms. 

Results for fixed-size ROIs are visualized in Table 2. For the freehand ROIs, ADC, D, D*, and pf did not show significant differences between the two groups. The test results for all four algorithms are shown in Appendix A.

## 4. Discussion

Recently, there has been an increased interest in using the IVIM-DWI technique in MRI to examine the correlation of IVIM parameters with specific molecular markers and to evaluate treatment responses. Applications of IVIM-DWI also include the characterization of both primary and metastatic tumors, such as liver cancer, thyroid nodules, and breast tumors [12,13,14,15]. The objective of the present work was to study the correlation between PD-L1 expression in NSCLCs and parameters extracted from IVIM-DWI MRI. We used a protocol to study the patients’ lungs with MRI that included a complete morphological assessment and an assessment of respiratory function, followed by the extraction of the IVIM-DWI features of lung tumors.

Our results suggest that IVIM-DWI can detect differences between no PD-L1 expression and positive PD-L1 expression in NSCLCs. Specifically, we observed that the mean D* value was statistically significant different between the two groups with the analysis of the fixed-size ROIs through algorithm 0. Moreover, we found a moderately negative correlation between mean D* and freehand segmented ROIs analyzed with algorithms 0 and 1, and between mean D* and fixed-size ROIs analyzed with algorithm 0. Looking at these results, our study confirms the good accuracy and high robustness of algorithm 0 and algorithm 1. The low accuracy of algorithm 2 may be explained by the fact that it ignores the contribution of low b values, which are instead used for computing the D* value. Algorithm 3 is characterized by a full non-linear fitting of all values, and this results in less robustness, which is an important parameter for lung image analysis. The better performance of fixed-size ROIs could be explained by the fact that they are placed on the tumor, thus avoiding the extreme periphery and the necrotic center, and may thus be more representative, which is different from the freehand ROIs which indiscriminately include all parts of the tumor.

Diffusion-weighted imaging reveals the motion of water molecules in biological tissues. The rate of diffusion in cellular tissues is described by the ADC, which reflects the microstructure of the investigated tissues [16]. Recently, ADC has been correlated with PD-L1 expression, and, specifically, Meyer et al. found a weak negative correlation between them in head and neck squamous cell carcinomas, probably due to the active metabolism and proliferation of the positive PD-L1 expression group, however the correlation was not strong enough to predict PD-L1 expression in clinical routine [17]. These results are in line with the ones of our study, in which the differences in ADC values between the positive PD-L1 expression and no PD-L1 expression groups in NSCLCs were not statistically significant. The D value is the pure diffusion coefficient, and it negatively correlates with tumor cellularity [18]. A meta-analysis published in 2020 summarizes the results of different studies focused on the diagnostic performance of IVIM-DWI-derived parameters in the differentiation of lung tumors, based on the tumor cellularity and perfusion characteristics. In particular, lung cancer resulted to have significantly lower ADC, D, and pf values compared to benign lesions, and the D value demonstrated the best diagnostic performance in the differential diagnosis of lung tumors [19]. In our study, the D value was not statistically different based on PD-L1 expression status, and this may reflect the fact that both groups of NSCLCs have high tumor cellularity. The pf value indicates the microcirculatory perfusion fraction, and it is generally positively related to the density of microvessels within the biological tissue in study, however, the correlation between pf and microvascular histology is not completely clear, and further studies are necessary to validate it [20]. The D* value is the perfusion-related diffusion coefficient, which indicates the diffusion effect caused by microcirculation perfusion in the local ROI and quantifies the collective motion of blood water molecules in the capillary network, flowing from one capillary segment to the next. Our study showed that the D* values were significantly decreased in the positive PD-L1 expression group compared to the group with no expression of PD-L1; this may be explained by the fact that infiltration of immune cells, which is greater in tumors with positive PD-L1 expression, is associated with the increased vessel permeability and the endothelial swelling, leads to a lower blood flow and stasis in local microcirculation [21]. In line with this finding, other studies have demonstrated the negative correlation between D* and inflammatory infiltrate, such as in the case of acute renal allografts rejection and for the characterization of inflammation in chronic liver disease [22,23].

Non-small cell lung cancers can be classified into three major immunophenotypes: (1) the immuno-inflammatory phenotype, characterized by an important infiltration of CD8+ T lymphocytes; (2) the immune-excluded phenotype, characterized by the production from the tumor and the surrounding microenvironment of signaling molecules such as TGF-β (transforming growth factor-beta), myeloid suppressor cells, and angiogenic factors, which prevent the immune cells from infiltrating the tumor; (3) the immune-desert phenotype, which is marked by poor invasion of CD8+ T lymphocytes and high tumor proliferation. It has been shown that there is a positive correlation between the number of inflammatory cells that infiltrate the tumor and the response to anti-PD-1/PD-L1 drugs, such as Nivolumab: indeed, the immunoinflammatory phenotype is the one that mostly responds to immunotherapy [24]. Moreover, Check-Mate 017 and Check-Mate 057 studies demonstrated that PD-L1 expression is positively associated with greater overall survival in patients treated with Nivolumab that have a NSCLC with ≥50% PD-L1 expression; however, an effective response is still observed in patients with ≥1% PD-L1 expression [25]. In patients with unresectable stage III NSCLC, a 5-year update of the PACIFIC study also confirmed PD-L1 expression as a biomarker for response [26]. Nonetheless, since PD-L1 expression is very heterogeneous in the tumor, it is difficult to accurately define the true PD-L1 status in the whole tumor using a single core biopsy [27,28,29], thus potentially explaining why only a fraction of patients treated with immunotherapy, between 20–40%, shows a stable and lasting response [30]. Still, we should consider that multiple endogenous and exogenous factors have an impact on the outcomes of immunotherapy in patients with NSCLC, acting as confounders; among these, epigenetic modulation, microbiome, obesity, and smoking history [31,32,33]. Of note, tobacco smoking, through its carcinogens, causes mutations in DNA, leading to the increased presence of neoantigens in tumors. The latter is responsible not only for the immunological recognition of the tumor, but also for the increment in PD-L1 expression on tumor cells, thus leading to increased responsiveness to immunotherapy [33,34].

In this context, it is necessary to have a more in-depth knowledge of the whole tumor microenvironment, and IVIM-DWI MRI could provide a practical benefit.

The most important advantage of IVIM-DWI MRI is that, as a non-contrast perfusion imaging modality, it can be used in situations in which intravenous administration of a contrast agent is not clinically justified or is contraindicated, such as in the case of patients with severely compromised renal function. Furthermore, IVIM-DWI MRI is non-invasive and does not involve ionizing radiation or injection of radioisotopes.

There are some limitations to this study. Firstly, it is a monocentric study, and the sample size is limited since the present data represents only a preliminary analysis of a bigger cohort study that is being progressively enrolled. Secondly, we used various algorithms for quantitative image analysis, which can be dispersive. Moreover, IVIM-DWI MRI of the lungs has important disadvantages including pulsation artifacts related to cardiac and respiratory motion; modest repeatability and reproducibility of its parameters, especially D*, probably due to their sensitivity to noise; and susceptibility of the sensitivity of IVIM-DWI MRI parameters depending on the utilized number and distribution of the applied b values. Moreover, IVIM-DWI-derived parameters are highly variable due to the lack of standardization of the IVIM-DWI technique between different centers, thus significant variance in calculated parameters among studies has been observed and no values for normal organs have been established [20].

In the future, IVIM-DWI parameters could offer the possibility to perform early diagnosis, pathological classification, and to assess responses to therapy. Elaborating an integrated standardization is necessary to obtain IVIM-DWI parameters broadly applicable in clinical practice and comparable between different centers.

## 5. Conclusions

Our preliminary results suggest that IVIM-DWI can detect differences among NSCLCs with different PD-L1 expression statuses. This could open up the possibility for the future development of IVIM-DWI MRI as a tool to biologically characterize NSCLCs.

## Figures and Tables

**Figure 1 cancers-14-05634-f001:**
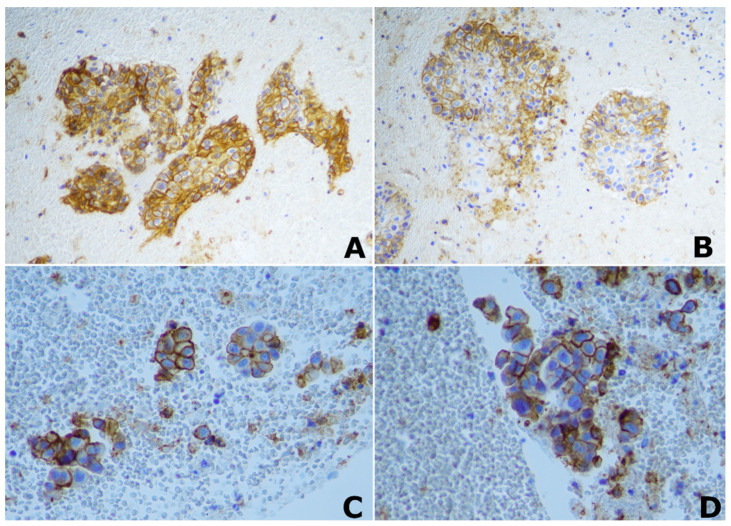
Immunohistochemical (IHC) staining of PD-L1. ×20 magnification image of positive PD-L1 IHC staining in two different patients with lung squamocellular carcinoma (**A**,**B**); ×40 magnification image of positive PD-L1 IHC staining in two different patients with lung adenocarcinoma (**C**,**D**).

**Figure 2 cancers-14-05634-f002:**
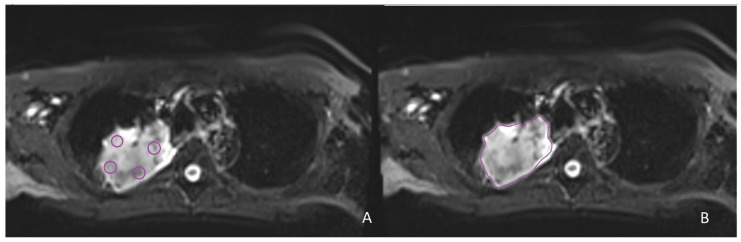
Examples of segmentations on intravoxel incoherent motion (IVIM) images in the same patient with non-small cell lung cancer localized in the right upper lobe. (**A**) Four fixed-size ROIs on the slice with the largest tumor diameter. (**B**) Freehand ROI on the slice with the largest tumor diameter.

**Figure 3 cancers-14-05634-f003:**
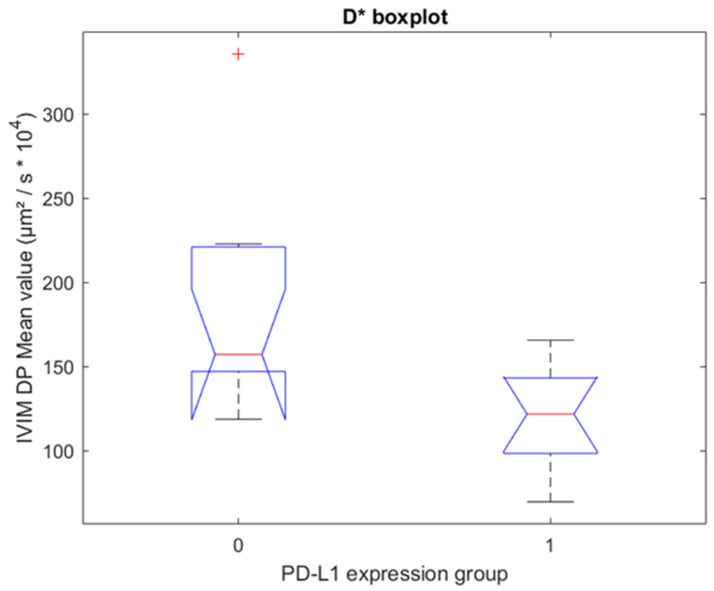
Boxplot for fixed-size ROIs. One boxplot for each group of PD-L1 expression (0 = no expression, 1 = positive expression), with the distribution of the 19 patients’ D* values for algorithm 0. Group 0 and Group 1 have statistically significant different median values (*p* = 0.008, Wilcoxon–Mann–Whitney test).

**Table 1 cancers-14-05634-t001:** Overview of the four prototype algorithms used in MR Body Diffusion Toolbox software to analyze the ROIs data. D: pure diffusion coefficient; D*: pseudodiffusion coefficient; pf: perfusion fraction.

Method	Summary
Algorithm 0	Linear fitting of D and pf, linear fitting of D*
Algorithm 1	Linear fitting of D and pf, non-linear fitting of D*
Algorithm 2	Linear fitting of D and pf, linear fitting of D*, ignoring D contribution for low b-values
Algorithm 3	Full non-linear fitting of D, pf and D*

**Table 2 cancers-14-05634-t002:** Results for fixed-size ROIs computed with algorithm 0. G0 refers to no expression of PD-L1 and G1 refers to the positive expression of PD-L1.

Results for Fixed-Size ROIs							
	Units	Mean		St. Dev.		*p*-Value WMW	Correlation with PD-L1 Protein
		G0	G1	G0	G1		
ADC	µm^2^/s × 10^6^	1445.8	1519.8	359.6	310.7	0.842	−0.073
D	µm^2^/s × 10^6^	1173.3	1327.0	289.5	373.0	0.356	0.035
D*	µm^2^/s × 10^4^	189.2	122.0	65.7	31.3	**0.008**	−0.374
pf	µm^2^/s × 10^3^	274.3	209.7	184.9	82.1	0.549	−0.148

## Data Availability

Not applicable. The data presented in this study are available in this article (and Appendix A).

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
