# Peer review of "Correlation between PD-L1 Expression of Non-Small Cell Lung Cancer and Data from IVIM-DWI Acquired during Magnetic Resonance of the Thorax: Preliminary Results"

_cancers, 2022, doi:10.3390/cancers14225634_

Round 1

Reviewer 1 Report

In present research, authors correlate the PD-L1 with IVIM-DWI status in stage III NSCLC. I have several reservations, my comments are appended a below:

1. My primary concern is that does present study distinguish between cytoplasmic and membrane PD-L1 as the latter one is known to play role in T cell anergy.

2. It may be risky to conclude base don limited size of patients. Authors should try to get retrospective data and support the findings.

3. Immunotherapy and PD-L1 is known to be affected by few cofounders like smoking, obesity. Authors should refer and discuss in brief.

4. Introduction- discuss details on prognosis observed on immunotherapy in lung cancer. Also, quote types of immunotherapies approved by FDA.

5. Author’s should provide evidence for IHC staining for PD-L1 in the studied patients.

6. Table 2- PD-L1 correlation- should clearly mention whether it is PD-L1 mRNA or protein.

Author Response

Dear Reviewer,  

We appreciate the constructive suggestions and the attentive comments you have made. We have incorporated most of them; changes are marked-up within the file “Revised Manuscript with Track Changes”.  

Our response follows.

1. My primary concern is that does present study distinguish between cytoplasmic and membrane PD-L1 as the latter one is known to play role in T cell anergy.

1. Thank you for your comment. In our study, in line with the general clinical practice, the extracellular domain of PD-L1 protein was quantified using the tumor proportion score (TPS), which is the percentage of viable tumor cells showing partial or complete membrane staining. Indeed, PD-L1 is a type I transmembrane protein containing an extracellular domain (ECD), a transmembrane domain (TMD), and a cytoplasmic domain (CD). Many studies have demonstrated that the ECD of PD-L1 binds to PD-1 on T cells to inhibit their tumor-killing activity and, analogously, the immunotherapeutic drugs which target the PD-1/PD-L1 axis bind the ECD of PD-L1. Four PD-L1 immunohistochemical assays registered with the FDA use four different ECD-targeting-PD-L1 antibodies on two IHC platforms (Dako and Ventana), each with a specified scoring system. On the other hand, the cytoplasmic domain of PD-L1 controls PD-L1 protein stability and degradation, but it is still not a target of immunotherapy.

2. It may be risky to conclude based on limited size of patients. Authors should try to get retrospective data and support the findings.

2. Thank you for your comment. Unfortunately, we are not able to provide retrospective data since we used the IVIM-DWI sequences in an experimental longitudinal setting; furthermore, the introduction of IVIM-DWI sequences even for experimental settings is quite recent and no retrospective data are available. We are aware of this limit, and we disclosed it in the Discussion. However, the ones presented are preliminary results of a cohort of patients that is progressively enrolled.

3. Immunotherapy and PD-L1 is known to be affected by few cofounders like smoking, obesity. Authors should refer and discuss in brief.

3. We added this in the Discussion (lines 307-310). Thank you for pointing this out.

4. Introduction- discuss details on prognosis observed on immunotherapy in lung cancer. Also, quote types of immunotherapies approved by FDA.

4. Thank you for your suggestion. We implemented the Introduction adding information on prognosis in patients with NSCLC treated with immunotherapy and types of immunotherapies approved by FDA (lines 61-82).

5. Author’s should provide evidence for IHC staining for PD-L1 in the studied patients.

5. We added in Materials and Methods the type of assay we used for the quantification of PD-L1 TPS and we included an image to show the IHC staining for PD-L1 (Figure 1).

6. Table 2- PD-L1 correlation- should clearly mention whether it is PD-L1 mRNA or protein.

6. We specified "Correlation with PD-L1 protein" in Table 2.

Reviewer 2 Report

Dear authors,

              I read this paper is very interesting for readers in the journal, and we can learn the possibility that IVIM-DWI MRI parameter D* could reflect PD-L1 expression in NSCLC. However, I would like to ask you describe the definition of ADC in detail in Materials and Methods, because many readers including me could be unfamiliar with the technical term. Finally, I agree with publication in cancers, thanks.

Author Response

Dear Reviewer,  

We appreciate the comments and constructive suggestions you have made. We added the definition of ADC and how it is calculated in Materials and Methods (lines 166-173).

Changes are marked-up within the file “Revised Manuscript with Track Changes”.  

Reviewer 3 Report

Minor Issues:

Line 149: patient's 

Please include the molecular mechanism of PD-L-1

Author Response

Dear Reviewer,  

We appreciate the constructive suggestions and the comments you have made. We have incorporated them in the article; changes are marked-up within the file “Revised Manuscript with Track Changes”.  

Our response follows. 

1. Line 149: patient's 

1. We corrected it.

2. Please include the molecular mechanism of PD-L-1

2. Thank you for your suggestion. We included the molecular mechanism of PD-L1 in the Introduction, as well as the prognosis in patients with NSCLC treated with immunotherapy (lines 61-82).

Round 2

Reviewer 1 Report

I recommend taking care of few more points: 

1. Point 3- I observe this part was missing in manuscript. I recommend to refer PMID: 33076303 and reiterate the role of cofounders affecting immunotherapy. 

2. Figure 3- indicate statistical inference. 

3. Figure 1- annotate with scale or mention magnification. 

Author Response

Dear Reviewer,

Thank you again for your comments. We have incorporated them; changes are marked-up within the file “Revised Manuscript with Track Changes v2”.  

Our response follows. 

1. Point 3- I observe this part was missing in manuscript. I recommend to refer PMID: 33076303 and reiterate the role of cofounders affecting immunotherapy.
1. We implemented the discussion on confounders affecting immunotherapy outcomes (lines 309-316) and we also added the reference you suggested to include.
2. Figure 3- indicate statistical inference.
2. We added statistical inference in the caption of Figure 3 (page 6): "Group 0 and Group 1 have statistically significant different median values (p = 0.008, Wilcoxon-Mann-Whitney test).".

3. Figure 1- annotate with scale or mention magnification.
3. We included magnification in the caption of Figure 1 (page 3): “Figure 1. Immunohistochemical (IHC) staining of PD-L1. x20 magnification image of positive PD-L1 IHC staining in two different patients with lung squamocellular carcinoma (a, b); x40 magnification image of positive PD-L1 IHC staining in two different patients with lung adenocarcinoma (c, d)”.